# A Novel Hemocyte-Derived Peptide and Its Possible Roles in Immune Response of *Ciona intestinalis* Type A

**DOI:** 10.3390/ijms25041979

**Published:** 2024-02-06

**Authors:** Shin Matsubara, Rin Iguchi, Michio Ogasawara, Hiroya Nakamura, Tatsuki R. Kataoka, Akira Shiraishi, Tomohiro Osugi, Tsuyoshi Kawada, Honoo Satake

**Affiliations:** 1Bioorganic Research Institute, Suntory Foundation for Life Sciences, 8-1-1 Seikadai, Seika-cho, Soraku-gun 619-0284, Kyoto, Japansatake@sunbor.or.jp (H.S.); 2Department of Biology, Graduate School of Science, Chiba University, 1-33 Yayoi-cho, Inage-ku 263-8522, Chiba, Japan; 3Department of Pathology, Iwate Medical University, 2-1-1 Idaidori, Yahaba-cho, Shiwa-gun 028-3695, Iwate, Japantrkata@iwate-med.ac.jp (T.R.K.)

**Keywords:** ascidian, *Ciona*, hemocyte, stomach, pharynx, peptide, immune response

## Abstract

A wide variety of bioactive peptides have been identified in the central nervous system and several peripheral tissues in the ascidian *Ciona intestinalis* type A (*Ciona robusta*). However, hemocyte endocrine peptides have yet to be explored. Here, we report a novel 14-amino-acid peptide, CiEMa, that is predominant in the granular hemocytes and unilocular refractile granulocytes of *Ciona*. RNA-seq and qRT-PCR revealed the high *CiEma* expression in the adult pharynx and stomach. Immunohistochemistry further revealed the highly concentrated CiEMa in the hemolymph of the pharynx and epithelial cells of the stomach, suggesting biological roles in the immune response. Notably, bacterial lipopolysaccharide stimulation of isolated hemocytes for 1–4 h resulted in 1.9- to 2.4-fold increased CiEMa secretion. Furthermore, CiEMa-stimulated pharynx exhibited mRNA upregulation of the growth factor (*Fgf3/7/10/22*), vanadium binding proteins (*CiVanabin1* and *CiVanabin3*), and forkhead and homeobox transcription factors (*Foxl2*, *Hox3*, and *Dbx*) but not antimicrobial peptides (*CrPap-a* and *CrMam-a*) or immune-related genes (*Tgfbtun3*, *Tnfa*, and *Il17-2*). Collectively, these results suggest that CiEMa plays roles in signal transduction involving tissue development or repair in the immune response, rather than in the direct regulation of immune response genes. The present study identified a novel *Ciona* hemocyte peptide, CiEMa, which paves the way for research on the biological roles of hemocyte peptides in chordates.

## 1. Introduction

An immune system against invading pathogenic microbes is essential for all living organisms [1,2,3,4]. The adaptive immune system is believed to be a slow but specific system acquired in jawed vertebrates; in contrast, the innate immune system is a rapid and primary defensive system against a broad spectrum of pathogens, not only in vertebrates but also in invertebrates [5,6]. The significant success of the adaptive immune system has been achieved by acquisition of vertebrate-specific molecules, including major histocompatibility complexes (MHCs), T-cell receptors (TCRs), and immunoglobulins (Igs) [1]. On the other hand, some invertebrates have evolved a wide variety of recognition (e.g., receptors of non-self antigens) and effector molecules (e.g., signaling molecules, complement system-related, antimicrobial peptides (AMPs), etc.) via genome-wide expansion or diversification of innate immune-related genes in a species-specific manner [7,8,9,10,11].

Ascidians are marine invertebrates belonging to the phylum Urochordata; these organisms represent the closest living relatives to vertebrates [12,13]. Given its phylogenetically important position in the superphylum Chordata, the cosmopolitan species *Ciona intestinalis* type A (synonymous with *Ciona robusta*) has been studied as a model organism in various fields of evolution, including genomics, developmental biology, endocrinology, and immunology [14,15,16,17,18,19,20,21,22]. The availability of genomic information and a growing body of RNA-seq data have facilitated studies of the molecular basis of the immune response in *Ciona* and the conservation of those in vertebrates [14,17,19,20,21,22,23,24,25].

The primary immune defense in ascidians is employed by the circulating hemocytes, the pharynx, and the alimentary canal [19,20,21,22]. Several genes involved in diverse pathogen recognition, including lectins, complements, and Toll-like receptors, have so far been demonstrated to be expressed in the *Ciona* hemocytes and pharynx [17,19,20,21,22]. In a previous study, thirty-four *Ciona* hemocyte-specific genes were identified, of which three were immune-related homologs of vertebrates (e.g., complement 6-like) [26]. However, twenty-three of the genes were either unknown or *Ciona*-specific, twelve of which were polypeptides (shorter than 100 residues). These suggest the presence of *Ciona*-specific roles of hemocyte peptides in immune response. Additionally, a genome-wide search for discovering AMPs identified numerous functionally uncharacterized peptides [27], implying novel biological roles for the unknown peptides. Nevertheless, large parts of these *Ciona*-specific peptides and their signaling in immune response remain to be elucidated.

In vertebrates, some bioactive peptides are known to regulate immune cells, while others show antimicrobial activities [28,29]. We have previously identified various neuropeptides in the *Ciona* neural complex [30] and demonstrated the reproductive function of several orthologous peptides to vertebrates [30,31,32]. In addition, we recently identified a novel *Ciona*-specific 51-amino-acid peptide, PEP51, and verified its possible roles in the activation of caspase in the ovary [33]. In *Ciona* hemocytes, several AMPs, including CrMAM-A, CrPAP-A, and others, have been reported [27,34,35]. Furthermore, a 73-amino-acid *Ciona* chemo-attractive peptide (CrCP) derived from an alternative transcript was upregulated in hemocytes by bacterial lipopolysaccharide (LPS) stimulation [36,37]. However, the endogenous roles of many other *Ciona*-specific hemocyte peptides remain largely unknown.

In the present study, we identified a novel 14-amino-acid peptide, CiEMa, from *Ciona* hemocytes. The objective of this research is to elucidate the biological roles of CiEMa.

## 2. Results

We previously performed peptidomic analysis on *Ciona* neural complexes and identified more than 30 neuropeptides [30]. In the neuropeptide-enriched gel-filtration fraction, MALDI-TOF mass spectrometry (MS) analysis identified several known neuropeptides (CiLF6, CiTKI, CiYFL1, and CiNTLP6) as well as an unknown major peak with an *m*/*z* of 1628.877, suggesting the presence of novel neuropeptide (Figure 1A). MS/MS analyses of this peak in the neural complex and ovary showed MS/MS patterns similar to those of the synthetic peptide of NERKGAEPQFPPEM-amide (Figure 1B). The MS-tag analyses for the neural complex detected most of the theoretical fragment peaks (Figure 1C) and confirmed the amino acid sequence that is encoded by the KY21.Chr12.349 gene (Appendix A, UniProt accession H2XSD0).

No BLASTP hits against the non-redundant protein database indicated that this peptide is *Ciona*-specific. Further in silico analyses using SignalP 6.0 and DeepLoc 2.0 for KY21.Chr12.349 predicted that the N-terminal sequences would be a signal peptide and the processed peptide is secreted (Appendix A). The aforementioned sequence of mature peptide is likely to be produced via an unusual cleavage between Trp and Asn at the N-terminus and Gly and Asp at the C-terminus, followed by amidation of the C-terminal Gly (Appendix A). We designated this novel peptide CiEMa based on the C-terminal sequence.

We then investigated the localization of CiEMa in the ovary and neural complex. Interestingly, in situ hybridization (ISH) of the ovary revealed that the *CiEma* mRNA was expressed specifically in interfollicular hemocyte cells but not in the ovarian follicle cells (Figure 2A,B). Consistent with the ISH results, immunohistochemistry (IHC) demonstrated that the CiEMa peptide was expressed predominantly in hemocytes (Figure 2C,D). These results confirmed the specificity of the ISH probes and anti-CiEMa antibodies. Similarly, in the neural complex, CiEMa was not expressed in the neural cells in the cerebral ganglion or neural gland (Figure 2E,F) but, rather, in a few hemocytes around the neural gland or in the ciliated funnel (Figure 2G,H).

Subsequently, we isolated *Ciona* hemocytes from the heart and investigated CiEMa expression. Seven to eight types of morphologically distinct *Ciona* hemocytes have been reported and classified into two groups: agranular and granular hemocytes [19]. Immunocytochemistry (ICC) demonstrated that no signals were detected in four types of agranular hemocytes (lymphocyte-like cells (LLCs), signet ring cells (SRCs), hyaline amoebocytes (HAs), or compartment cells (CCs)). On the other hand, in granular cells, weak and strong signals were observed in GHs and URGs, respectively, but not in morula cells (MCs) (Figure 3). These results confirmed the hemocyte expression of CiEMa and suggested its roles in the immune system.

We referred to the *CiEma* expression during early embryo development and found that *CiEma* was first expressed at around the juvenile stage [38]. We then examined the expression in two-week-old juveniles by whole-mount ISH (WISH). As in Figure 2A,B, specific signals were observed using antisense probes (Figure 4A–D) and not with the control (sense) probes (Figure 4E–H). In accordance with Figure 2 and Figure 3, the *CiEma* mRNA expression in juveniles was observed predominantly in the hemocytes distributed in the neural complex (Figure 4B,F), pharynx (Figure 4C,G), and stomach (Figure 4D,H). Moreover, distinct broad signals were also observed in the stomach (Figure 4D,H).

We previously obtained transcriptomic data from 11 samples of 9 adult tissues (Figure 5A) [18], leading to the detection of high *CiEma* expression in the pharynx and stomach but low in the neural complex and ovary (Figure 5B). The tissue distribution and statistically significant expression in the pharynx were confirmed by qRT-PCR (Figure 5B). Furthermore, IHC confirmed the specific expression of CiEMa in the hemocytes of the pharynx and stomach (Figure 6). Of note, some parts of the body fluid (hemolymph) showed strong signals (Figure 6A, arrowheads), indicating the secretion of CiEMa by hemocytes. The epithelial cells of the inner fold also showed strong signals that were not observed in the outer fold of the stomach. Furthermore, both the apical and basolateral sides of the epicardium were strongly stained (Figure 6B), suggesting multiple roles of CiEMa in the *Ciona* stomach.

Subsequently, we investigated the effects of a bacterial LPS as a non-self-antigen on CiEMa expression, given that *Ciona* pharynx and stomach are most exposed to the microbiome in the marine environment and are the major organs of immunity [19]. We treated isolated hemocytes with LPS and examined the mRNA expression and secretion of CiEMa. Although some AMP genes have been reported to be upregulated in 1 h after LPS challenge [27], no significant change was observed in *CiEma* mRNA levels following LPS stimulation (Figure 7A). Next, CiEMa secretion was examined by dot blot analyses using supernatants of hemocytes that had been incubated in the absence or presence of LPS. In contrast to the case with mRNA, stronger signals were observed at every time point following LPS stimulation (1, 2, and 4 h) (Figure 7B, left). Quantification using Fiji software (ver. 1.54f) demonstrated that those signals were increased significantly 2.7-, 1.9-, and 2.4-fold at 1, 2, and 4 h, respectively (Figure 7B, right). The original images and quantification file are provided as Appendix A and Appendix A. These results indicated that the secretion of CiEMa by hemocytes increases in response to stimulation with LPS, a non-self-antigen.

Given the strong signals in the hemolymph (Figure 6A), the pharynx is presumed to be the primary target of CiEMa. Therefore, we isolated the pharynx, stimulated it with 1 μM CiEMa for 0, 1, 2, 4, and 8 h, and subjected it to RNA-seq. The resulting reads, mapping rates, and NCBI SRA accessions are summarized in Appendix A. The expression level (TPM, transcripts per million) of each gene was calculated and is provided in Appendix A. The expression patterns following CiEMa stimulation were confirmed by qRT-PCR (Figure 8). Unexpectedly, none of the known *Ciona* AMP genes, including *CrPap-a*, *CrMam-a*, KY21.Chr7.694 (KH.C7.94), and KY21.Chr2.890 (KH.S908.1), exhibited significant changes in response to CiEMa (Figure 8A–D). A time-dependent change in the immune-related genes in the *Ciona* pharynx during LPS challenge has been reported [19]. Among the genes, *Tgfbtun3* was significantly downregulated (Figure 8E), while *Mmp2/9/13* was upregulated 8 h after CiEMa stimulation (Figure 8H). The cytokine expressions (*Tnfa* and *Il17-2*) showed no significant change during CiEMa stimulation (Figure 8F,G). Of particular interest is that a few signaling genes including *Ghsr-like* (6.7-fold) and the growth factor, *Fgf3/7/10/22* (6.4-fold), and several forkhead and homeobox transcription factors (*Foxl2* (3.7-fold), *Hox3* (4.4-fold), *Dbx* (18.6-fold), and *Prrx* (6.4-fold)) were upregulated by CiEMa (Figure 8I,J,M–P). Moreover, vanadium-binding proteins (*CiVanabin1* (5.9-fold) and *CiVanabin3* (7.2-fold)) were also significantly induced (Figure 8K,L). These results demonstrated that CiEMa induces various gene expressions in the pharynx.

## 3. Discussion

In the past two decades, along with the assembly of the genome and cDNA libraries, various *Ciona* hemocyte-derived transcripts have been identified and attracted attention to the evolutionary lineage of the innate immune system [23,26,40,41]. Based on genome sequencing data and homology searching, several immune-related genes, including complement components, Toll-like receptors, lectins, and cytokines, have been characterized as counterparts of their vertebrate homologs [17,19,23]. The easy isolation and fractionation methods of *Ciona* hemocytes, and the application of LPS challenge to ascidian individuals, enabled us to investigate the immune system of ascidians and underscored the usefulness of *Ciona* as an evolutionary model organism of the innate immune system [19]. In contrast to the homologous molecules to vertebrates, *Ciona*-specific molecules have been less investigated. In this study, we identified a novel hemocyte-derived peptide, CiEMa, and demonstrated the possible roles in the immune response of the *Ciona* pharynx.

We first identified CiEMa from the neural complex and ovary and found it to be expressed in the specific hemocytes and stomach. In the stomach, the epithelial cells, especially those of the inner fold, are known to express several genes involved in pinocytosis and phagocytosis [42]. Particularly, the strong expression of CiEMa in the bottom of inner fold cells was similar to that of the phagocytosis-related cell surface receptor Mrc1 (mannose receptor C-type 1). The endosome-like expression of CiEMa in the stomach epithelium supports a possible role in the uptake of large particles and/or small nutrients. Moreover, the variable region-containing chitin-binding protein A (VCBP-A) has been shown to accumulate in identical large (endosome-like) vacuoles in the inner fold [43], raising the possibility that CiEMa supports the opsonizing function of VCBP-A. In contrast, CiEMa is unlikely to be involved in the digestion, given that it is not expressed in the outer fold cells where it is enriched for pancreatic enzymes [44]. Thus, CiEMa may play distinct roles in the epithelial cells of the stomach and in hemocytes.

Consistent with the fact that hemocytes initially emerge at the first ascidian stages of juveniles [45], CiEMa showed the most predominant expression from juvenile to adult hemocytes. Of interest, the CiEMa signal was not observed in all hemocytes but only in some clusters of GH and URG hemocytes, suggesting specific roles in these particular cell types. GHs and URGs have been reported to express various immune-related genes, including cytokines, phenoloxidases, and complements [19,20,22]. Although *CiEma* mRNA expression did not change following LPS stimulation, increased CiEMa secretion from isolated hemocytes was observed. Similar expressions have been reported for some *Ciona* cytokines and relevant signaling molecules, including CiTGFβ, CiTNFα, and CiIL17s, in the clustered hemocytes of LPS-challenged ascidians [46,47,48], supporting the view that CiEMa plays a cytokine-like role as a signal transduction molecule in response to non-self-antigens. Identification of the CiEMa receptor will be useful for elucidating biological roles in other tissues (e.g., neural complex and ovary). Response to other non-self-antigens (e.g., zymosan, poly (I:C), and flagellin) is also of interest.

Most of the known hemocyte-derived signaling molecules are cytokines [19,20,22]. The three *CiIl17* genes have been shown to upregulate following LPS challenge [47]. The *CiTnfa* has also been shown to be induced in hemocytes 4 h after LPS injection [46]. Of particular interest is that CiEMa did not alter the *Tnfa* expression and downregulated *Il17-2*, suggesting that CiEMa plays roles distinct from those of typical cytokines. Moreover, none of the examined AMP genes were affected by CiEMa stimulation, suggesting that CiEMa is likely to be involved in other functions rather than direct regulation of immune processes. 

In general, FGF signaling, homeobox proteins, and forkhead-box proteins are key regulators of cell proliferation, differentiation, and embryonic development in both vertebrates and invertebrates. *Ciona* FGF3/7/10/22 is important for notochord development in tailbud-stage embryos [49]. In contrast, disruption of *CiHox3* did not affect the normal expression of neuronal markers in swimming larva [50]. Although no other functional insights of signaling genes and transcription factors in Figure 8 in adult tissues have been reported, upregulation in 4 or 8 h after CiEMa stimulation implies possible roles in tissue development or repair via cell proliferation and differentiation. In addition, five vanadium-binding protein genes, *CiVanabins*, have been identified [51], two of which, *CiVanabin1* and *CiVanabin3*, were upregulated by CiEMa. In *Ascidia sydneiensis samea*, vanabins have been shown to be expressed in SRCs, a type of agranular hemocyte [52,53]. In *C. intestinalis* type A, SRCs have been shown to express the AMP CrMAM-A and the galectins (CrGal-a and CrGal-b) [19,54,55]. These findings raise the possibility that vanadium accumulation contributes to the immune response in ascidians. Consequently, the current results strongly suggest that CiEMa plays a role in cell growth and/or tissue repair via regulation of the growth factor and transcription factors, rather than direct regulation of immune response genes including AMPs and cytokines. Taken together, the present study verified the novel cascade of immune response mediated by CiEMa; the non-self-antigen LPS acts on GHs and URGs to induce CiEMa secretion, which leads, in turn, to the upregulation of various genes in the pharynx (Figure 9). Given that most of the known cytokines are proteins, identification of the hemocyte-derived peptide and its unconventional signaling of immune response will contribute to the comprehensive understanding of the immune system of *Ciona*.

Most vertebrates have acquired sophisticated adaptive immune systems, employing MHCs, TCRs, and Igs. These changes are hypothesized to have led to decreases in the number of components in the innate immune system [1]. In contrast, in invertebrates, various innate immune molecules have evolved, such as lectins [56], AMPs [57], and cytokines [58]. Additionally, *Ciona* has developed a variety of species-specific peptides, including CrCP [36,37], PEP51 [33], and CiEMa. As observed in CiEMa, a signaling molecule produced by atypical processing at Trp or Asp may contribute to the diversification of signal transduction in the ascidian immune response. Combined with the fact that such unconventional processing of peptides has also been reported in other invertebrates [59,60,61,62], the observation of numerous functionally uncharacterized peptides in *Ciona* suggests that these factors serve as novel cytokine-like or immune-related signaling peptides. Although further functional analyses are required, the current study identified a novel hemocyte-derived peptide and proposed possible roles in the immune response in *Ciona*, which paves the way for the research of the biological roles of hemocyte peptides in chordates.

## 4. Materials and Methods

### 4.1. Animals

Adult ascidians (*Ciona intestinalis* type A, *Ciona robusta*) were cultivated at the Maizuru Fisheries Research Station of Kyoto University or the Misaki Marine Biological Station of the University of Tokyo, where the animals were maintained at 18 °C in sterile artificial sea water (ASW). Two-week-old juveniles were produced by artificial insemination, as reported previously [44].

### 4.2. Peptide Extraction and Detection by Mass Spectrometry

Peptides from adult *Ciona* tissues were extracted and purified as previously [30]. In brief, 20 neural complexes and 5 ovaries were homogenized in liquid nitrogen. Peptides were obtained using an extraction buffer (methanol/water/acetic acid, 90:9:1) supplemented with PMSF. Following incubation for 30 min at room temperature, the buffer was exchanged for 30% acetonitrile containing 0.1% trifluoroacetic acid (TFA) using a Speed-Vac lyophilizer (Sakuma, Tokyo, Japan). The peptide-enriched fraction from the neural complex was then separated by Superdex 30 Increase 10/300 GL gel filtration column chromatography (10 × 300 mm, GE Healthcare, Buckinghamshire, UK). The eluent was dried using a Speed-Vac, and the precipitates (peptides) were dissolved in 0.1% TFA. Separately, the predicted mature peptide (NERKGAEPQFPPEM-amide) was synthesized and purified commercially (PH Japan Co., Ltd., Hiroshima, Japan). The peptides were analyzed using a rapifleX MALDI-TOF spectrometer (Bruker Daltonics, Bremen, Germany). The MS/MS spectrum data for the peptide extracts from the neural complex were analyzed using MS-Tag (https://prospector.ucsf.edu/prospector/cgi-bin/msform.cgi?form=mstagstandard, accessed on August 2023) for confirmation of the amino acid sequences. The protein sequences including KY21.Chr12.349 were obtained from the *Ciona* ghost database (http://ghost.zool.kyoto-u.ac.jp/default_ht.html, accessed on August 2023) [63]. Analyses were conducted as more than three independent experiments.

### 4.3. In Situ Hybridization (ISH)

mRNA localization in the *Ciona* ovary was performed as previously [31]. In brief, cDNA fragments of CiEMa (KY21.Chr12.349) were obtained from *Ciona* ovaries using a primer pair (ACGCATTCCAGACAAATCTCAA and GCTCCAATGATCCTTTGCAGC) and cloned into the pCR™4-TOPO Vector (Thermo Fisher Scientific, Waltham, MA, USA). The sequence-confirmed vector was linearized using *Not*I or *Pme*I and used for digoxigenin (DIG) labeling (Roche, Basel, Switzerland). Adult *Ciona* ovaries were fixed overnight at 4 °C with 4% paraformaldehyde (PFA) in ASW; the fixative was exchanged with 30% sucrose, and the tissue was embedded in super cryoembedding medium (SCEM). Further, 10 μm cryosections were prepared and subjected to ISH as previously [31]. Specifically, hybridization was performed at 60 °C for 16 h in a hybridization buffer (50% formamide, 10 mM Tris-HCl, 1 mM EDTA, 0.6 M NaCl, 10% dextran sulfate, 1 × Denhardt’s solution, 0.25% SDS, and 0.2 mg/mL yeast transfer RNA). After washing and blocking of the slides, signals were developed using alkaline phosphatase-conjugated anti-DIG antibody (Roche, 1:5000) and NBT/BCIP (Nacalai Tesque, Inc., Kyoto, Japan) system. Three independent ovaries were examined. For the whole-mount experiment, two-week-old *Ciona* juveniles were fixed with 4% PFA and whole-mount ISH (WISH) was performed using the “InSitu Chip”, as previously described [26].

### 4.4. Antibody Generation and Purification

The CiEMa-antigen peptide (KLH-CNERKGAEPQFPPEM-amide) was synthesized and purified commercially (PH Japan Co., Ltd.). A CiEMa-specific rabbit antibody was raised by immunization of two animals (Eurofins Genomics, Tokyo, Japan). The anti-CiEMa antibodies were affinity purified using KLH-free antigen peptides (Eurofins Genomics). The purified antibodies were stocked at 0.29 mg/mL in a solution of 50% glycerol in phosphate-buffered saline (PBS) and maintained at −30 °C.

### 4.5. Immunohistochemistry (IHC)

Adult *Ciona* tissues (ovary, neural complex, pharynx, and stomach) were fixed overnight at 4 °C with Bouin’s solution; following exchange of the fixative for 30% sucrose in PBS, the ovaries were embedded in SCEM and cryosectioned at 10 μm. IHC was performed as previously [31]. The affinity-purified anti-CiEMa antibody, diluted 1:150–300 in Can Get Signal immunostain Solution A (Toyobo, Osaka, Japan), was used as the primary antibody. To generate the pre-absorbed antibody for a negative control, the anti-CiEMa antibodies were preincubated with an excess (approximately 80 μg) of KLH-free CiEMa. Immunoreactivity was developed using an Avidin-Biotin Complex (ABC) kit (Vector Laboratories, Burlingame, CA, USA) according to the instructions. Two independent tissues were examined.

### 4.6. Hemocyte Collection and Immunocytochemistry (ICC)

Adult *Ciona* hemocytes were collected by rupturing the heart, and the cells were subjected to centrifugation (1000× *g*, 5 min, 18 °C). Both the supernatant and pellet were used in subsequent experiments. The supernatant was filtered and used as hemolymph. The cell pellet was suspended in an anticoagulant buffer (11 mM KCl, 43 mM Tris-HCl, 0.4 M NaCl, 10 mM EDTA) [64] and used as hemocytes. Aliquots of the hemocyte suspension were placed on slides allowed to dry at room temperature, then subjected to ICC. In brief, the slides were further dried for 1 h at 37 °C before fixing for 10 min with 4% PFA in PBS. Subsequent steps of blocking, immunoreaction, and signal detection were performed as for IHC. The experiment was performed as three independent analyses.

### 4.7. RNA-Seq Data Analysis

RNA-seq data of the adult tissues were obtained previously [18]. The reads were mapped to the *Ciona* genome (KY21 Gene Model) [65], using HISAT2 (version 2.1.0) [66] and quantified using RSEM (version 1.3.3) [67]. The resulting gene expression levels are presented as exported values of transcripts per million (TPM).

### 4.8. RNA Purification and qRT-PCR

Adult ascidian tissues (oral siphon, atrial siphon, neural complex, endostyle, heart, ovary, pharynx, stomach, and intestine) were collected as previously [18]. Total RNA from adult tissues and hemocytes was extracted, purified, and depleted of genomic DNA as previously [18]. An aliquot of 1 mg (for tissues) or 200 ng (for hemocytes) of DNase-treated total RNA was used for first-strand cDNA synthesis. qRT-PCR was performed using a CFX96 Real-time System and SsoAdvanced™ Universal SYBR Green Supermix (Bio-Rad Laboratories, Hercules, CA, USA). The primers are listed in Appendix A. Gene expression levels were normalized to the reference genes: KY21.Chr10.446 for Figure 5 [18]; KY21.Chr9.158 (KH.C9.410) for Figure 7A [27]; and KY21.Chr2.148 for Figure 8. The KY21.Chr2.148 was identified by RNA-seq below to be constantly expressed following CiEMa stimulation. Given that the expression of the AMP genes at 0 h varied among the datasets, the expression levels were normalized to the 0 h values and set as 1. Three to four independent sets of samples were examined.

### 4.9. LPS Stimulation

Hemocytes were collected from 2 to 3 ascidians, as described above, and the cell number was quantified using a CDA-1000 particle counter (Sysmex Corporation, Hyogo, Japan). Cells were seeded at 1 × 10^6^–2 × 10^6^/well in a 24-well plate in medium consisting of sterile seawater (Nazeme 800; Japan QCE, Numazu, Japan) containing 20% hemolymph and 0.5× penicillin and streptomycin (Nacalai Tesque, Inc.). The plates were incubated for 16 h at 18 °C. The hemocytes were then stimulated without or with LPS from *E. coli* 055:B5 (0.1 mg/mL, Millipore/Sigma-Aldrich, St. Louis, MO, USA) for 1, 2, and 4 h at 18 °C. Four to six independent sets of the hemocytes (1 h) and culture supernatants (1, 2, and 4 h) were collected and used for qRT-PCR and dot blot analyses, respectively.

### 4.10. Dot Blot Analyses

The culture supernatants of hemocytes (incubated with or without LPS for 1, 2, and 4 h) were loaded onto a Sep-Pak C18 1 cc Vac Cartridge (Waters Corporation, Milford, MA, USA), washed with 10% acetonitrile, and eluted with 30% acetonitrile containing 0.1% TFA. For each sample, the eluent was dried using a Speed-Vac and dissolved in 10 mL of water. Then, 1 mL of each sample was spotted onto a nitrocellulose membrane. The membrane was dried for 30 min at 100 °C, then blocked for 30 min with Block Ace (KAC Co., Ltd., Kyoto, Japan) in TBS containing 0.05% Tween 20 (TBST). The membrane was incubated with the affinity-purified anti-CiEMa antibody (1:2000), followed by the HRP-conjugated anti-rabbit IgG secondary antibody (1:2000, GE Healthcare, Buckinghamshire, UK). Signals were developed using an ECL substrate (GE Healthcare). Six different sets of samples were analyzed. Signals were captured using an Amersham™ Imager 600 (GE Healthcare) and then quantified using Fiji software [68]. The signals of pre-absorbed control were subtracted for CiEMa quantification.

### 4.11. CiEMa Stimulation and RNA Sequencing

The isolated pharynx was incubated with 1 mM synthetic CiEMa in sterile seawater (Nazeme 800) containing 0.5 × penicillin and streptomycin (Nacalai Tesque, Inc.) for 0, 1, 2, 4, and 8 h at 18 °C. Total RNA was extracted, purified, and depleted of genomic DNA, as described above. One and four sets of samples were collected and used for RNA-seq and qRT-PCR, respectively. An aliquot (500 ng) of quality-confirmed RNA was used for library construction and sequenced at Novagene (Beijing, China) using the Illumina NovaSeq 6000 platform (Illumina, San Diego, CA, USA). The resulting fastq files were analyzed as described above. The expression level (TPM) of each gene is listed in Appendix A. The fastq files were deposited in the NCBI database (Accession No. PRJNA1045840). The RNA-seq data were confirmed by qRT-PCR as described above.

### 4.12. Statistical Analysis

Statistical analyses were performed using R software (version 4.2.2), as previously described [18]. In brief, the Levene test was initially performed to confirm the equal variance of expression level in each tissue (Figure 5B). Subsequently, the data were analyzed by a parametric one-way Analysis of Variance (ANOVA) with post hoc Tukey’s multiple comparison tests (Figure 5B). For Figure 7 and Figure 8, comparisons were conducted using a non-paired Student’s *t*-test and one-way ANOVA followed by post hoc tests, respectively. Note that, in Figure 8, the one-way ANOVA for each AMP gene was performed while excluding data from the 0 h time point. Where applicable, analyses were performed as two-tailed tests. *p* < 0.05 was considered statistically significant. *p*-values for the Levene test, ANOVA, and Student’s *t*-test are indicated as *P_L_*, *P_A_*, and *P*, respectively.

## Figures and Tables

**Figure 1 ijms-25-01979-f001:**
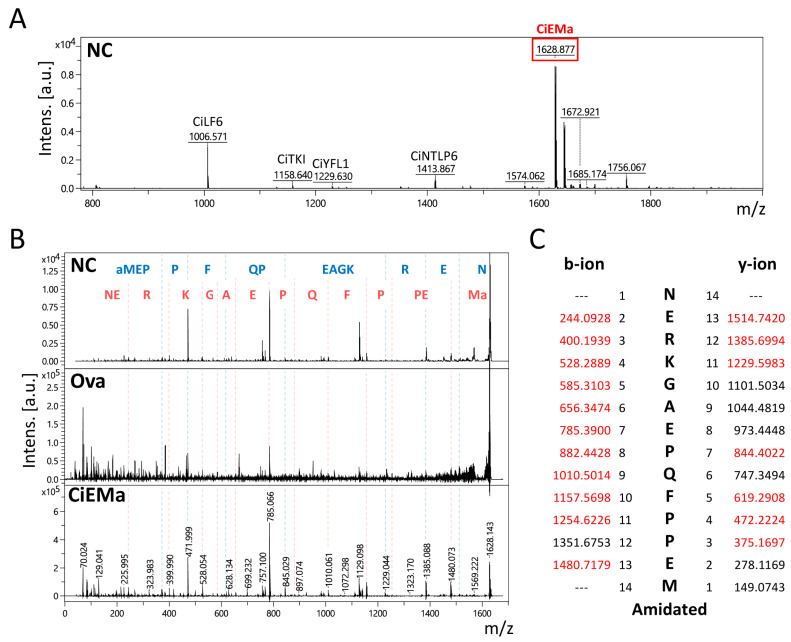
Identification of a novel peptide, CiEMa, from the *Ciona* neural complex and ovary. (**A**) MALDI-TOF analysis of the peptide-enriched fraction of the neural complex. Several annotated peptides, including CiLF6, CiTKI, CiYFL1, and CiNTLP6, were detected. A predominant peak with an *m*/*z* of 1628.877 was also observed. (**B**) Tandem MS analyses on the precursor ion of 1628.5 in the neural complex and ovary. Most of the MS/MS peaks of the neural complex and ovary were identical to those of a synthetic peptide (NERKGAEPQFPPEM-amide). The peptide sequences that were identified by b- (red) and y-ions (blue) are shown at the top. NC, neural complex; Ova, ovary. (**C**) MS-tag analyses, using a mass list of fragment ions of the neural complex, identified theoretical b- and y-ions (shown in red). The experiments were performed independently at least three times.

**Figure 2 ijms-25-01979-f002:**
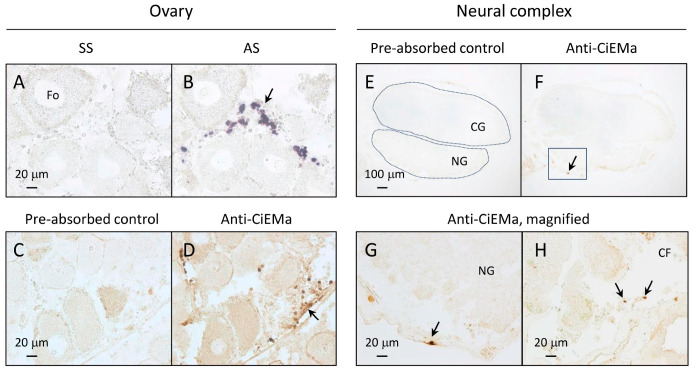
Localization of CiEMa in the ovary and neural complex. (**A**–**D**) In situ hybridization (ISH, (**A**,**B**)) and immunohistochemistry (IHC, (**C**,**D**)) of CiEMa in the ovary. 4% PFA- and Bouin’s-fixed ovaries were used for the ISH and IHC, respectively. Signals were detected using the DIG-NBT/BCIP system for ISH and the ABC system for IHC, respectively. For ISH, the sense probe was used as a negative control; specific signals were observed in the antisense probe (arrow). For IHC, the pre-absorbed antibody was used as a negative control; specific signals were observed in the anti-CiEMa antibody (arrow). Fo, follicles. The scale bar represents 20 μm. (**E**–**H**) IHC of CiEMa in the neural complex. (**E**,**F**) Low-magnification images indicated no signals in the cerebral ganglion (CG) and neural gland (NG). The boxed area in the upper panel is magnified and shown below (**G**). (**G**,**H**) High-magnification images indicate specific expression in the hemocytes around the NG and in the ciliated funnel (CF) (arrows). Scale bars in the (**E**–**H**) represent 100 μm and 20 μm, respectively. Localization was confirmed using two or three independent tissues.

**Figure 3 ijms-25-01979-f003:**
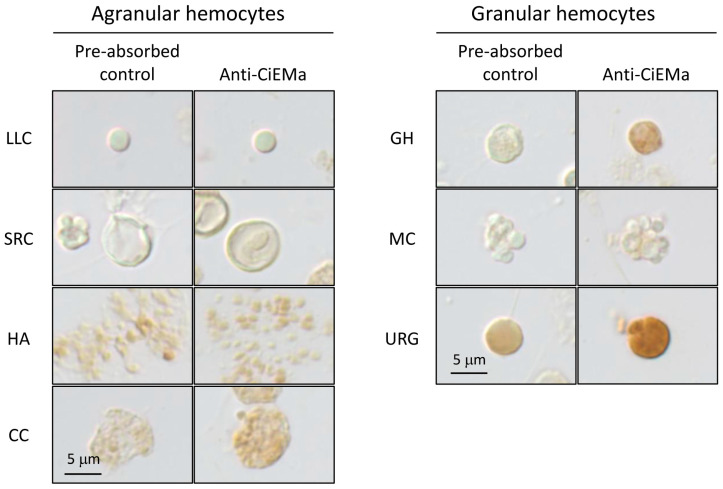
Localization of CiEMa in *Ciona* hemocytes. *Ciona* hemocytes were isolated by rupturing the heart, collected in an anticoagulant-containing buffer, dried on the slides, and subjected to immunocytochemistry (ICC). Blocking, immunoreaction, and signal detection were performed as IHC. The *Ciona* hemocytes were distinguished by their morphology. LLC, lymphocyte-like cell; SRC, signet ring cell; HA, hyaline amoebocyte; CC, compartment cell; GH, granular hemocyte; MC, morula cell; URG, unilocular refractile granulocyte. The scale bar represents 5 μm. Expression was confirmed in three independent experiments.

**Figure 4 ijms-25-01979-f004:**
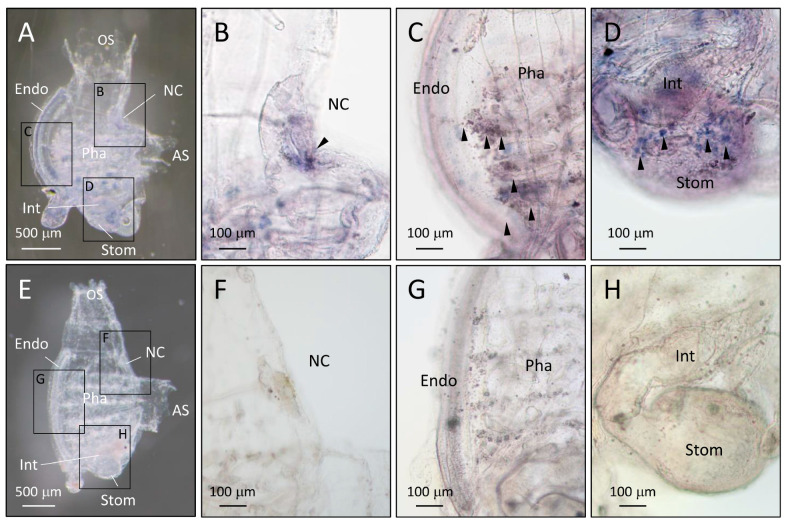
Localization of *CiEma* in *Ciona* juveniles. Two-week-old juveniles were used for WISH. Panels (**A**,**E**) show overviews of *Ciona* juveniles at low magnification following hybridization with antisense (**A**) and sense (**E**) probes. The indicated areas in (**A**,**E**) are shown at high magnification in (**B**–**D**) and (**F**–**H**), respectively. The hemocyte-specific signals (arrowheads) were observed in the neural complex (**B**,**F**), pharynx (**C**,**G**), and stomach (**D**,**H**). The stomach also showed broad signals (**D**,**H**). OS, oral siphon; AS, atrial siphon; NC, neural complex; Pha, pharynx; Stom, stomach; Int, intestine; Endo, endostyle. Scale bars represent 500 μm in (**A**,**E**) and 100 μm in (**B**–**D**) and (**F**–**H**), respectively. Expression was confirmed using approximately ten juveniles.

**Figure 5 ijms-25-01979-f005:**
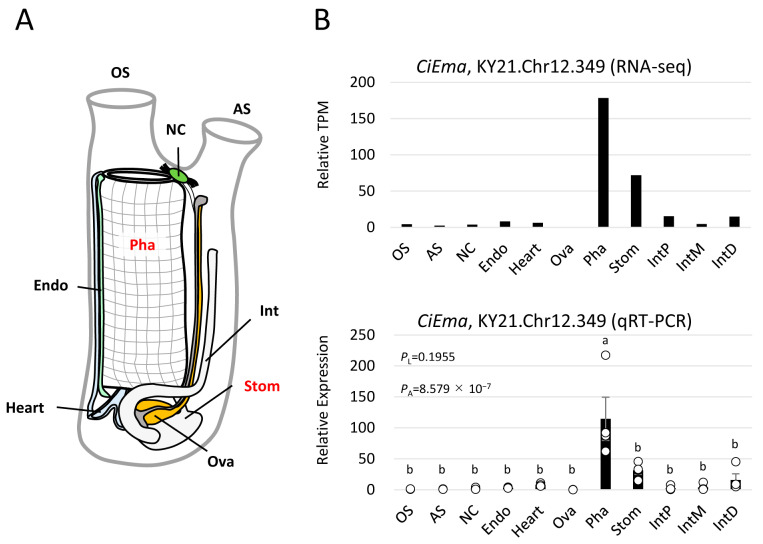
Tissue distribution of *CiEma* mRNA in adult *Ciona*. (**A**) A schematic illustration of adult *Ciona* tissues (modified from Osugi et al., 2020 [39]). OS, oral siphon; AS, atrial siphon; NC, neural complex; Pha, pharynx; Stom, stomach; Int, intestine; Endo, endostyle. (**B**) The previous RNA-seq data [18] for adult *Ciona* tissues were analyzed (**upper**). qRT-PCR analysis confirmed the tissue distribution of *CiEma* (**lower**). Relative expression to the reference gene (KY21.Chr10.446) [18] was indicated. Data are shown as the mean ± SEM with data points. Four independent data sets were analyzed using the Levene (*P_L_* = 0.1955) test followed by one-way ANOVA (*P_A_* = 8.579 × 10^−7^). Different alphabets are considered statistically significant (*p* < 0.05). IntP, proximal intestine; IntM, middle intestine; IntD, distal intestine.

**Figure 6 ijms-25-01979-f006:**
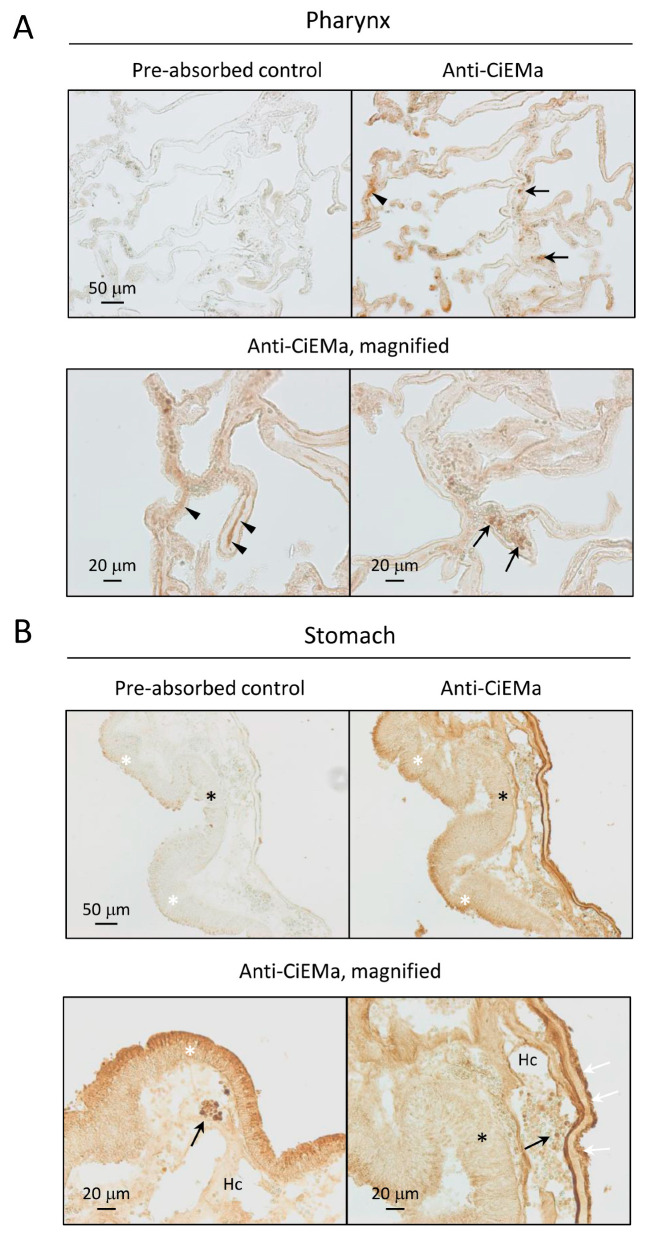
Localization of CiEMa in the pharynx and stomach. The Bouin’s-fixed pharynx (**A**) and stomach (**B**) were used for IHC. Serial sections were probed with either the pre-absorbed negative control or the anti-CiEMa antibody. (**A**) Signal was detected in the hemocytes (black arrows) and hemolymph (arrowheads). (**B**) In addition to the hemocytes (black arrows), strong signals were detected in the epithelial cells of the inner fold (white asterisk) but not in the outer fold (black asterisk). Strong signal was observed on both the apical and basolateral sides of the epicardium (white arrows). Hc, hemocoel. Scale bars in the upper and lower panels represent 50 μm and 20 μm, respectively. Expression was confirmed in two different tissues.

**Figure 7 ijms-25-01979-f007:**
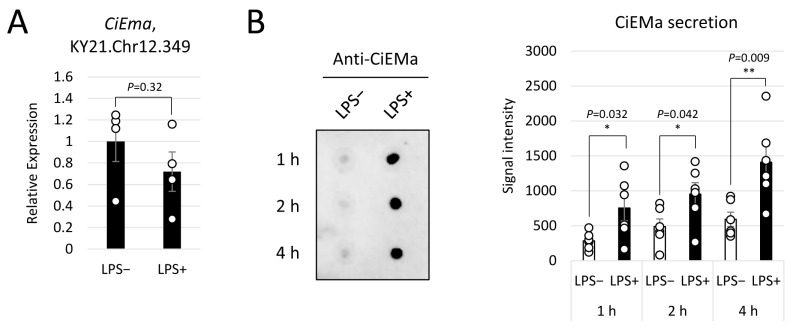
Induction of CiEMa secretion by LPS challenge. (**A**) Hemocytes isolated from adults were incubated with or without LPS for 1 h, and *CiEma* expression was examined by qRT-PCR. Relative expression to the reference gene (KY21.Chr9.158 (KH.C9.410)) [27] was indicated. Four independent data sets were analyzed by Student’s *t*-test (*p* = 0.32) and are presented as the mean ± SEM with data points. (**B**) Hemocytes were incubated with or without LPS for 1, 2, and 4 h, and CiEMa secretion was examined by dot blot. Signals were quantified using Fiji software. Data are shown as the mean ± SEM with data points. Six independent data sets were analyzed by Student’s *t*-test *, *p* = 0.032, 0.042, and **, 0.009 for 1, 2, and 4 h, respectively.

**Figure 8 ijms-25-01979-f008:**
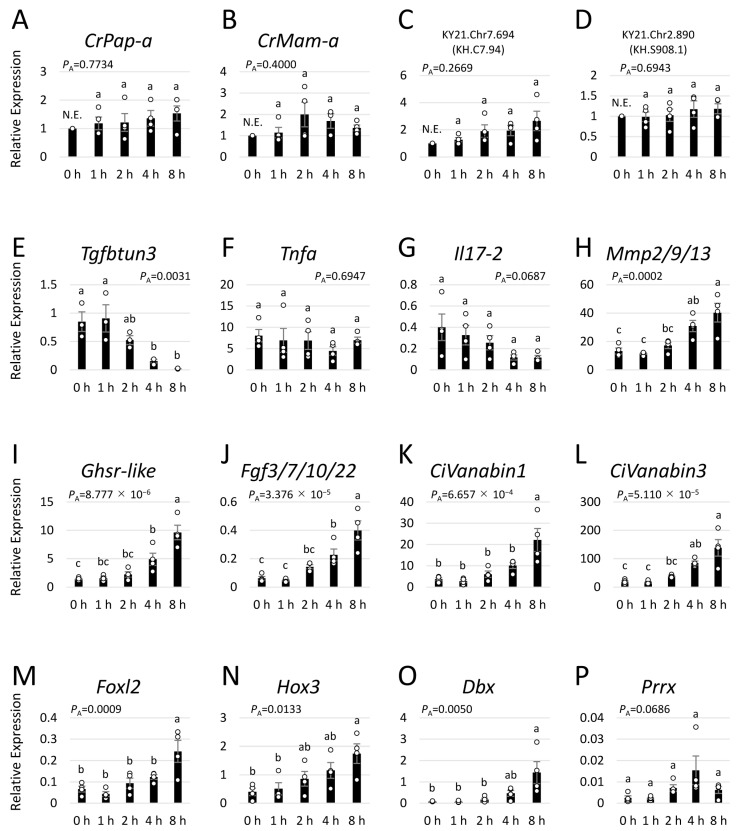
Gene expression change in the CiEMa-stimulated pharynx. Differentially expressed genes screened on RNA-seq data were confirmed by qRT-PCR. Expressions of the antimicrobial peptide genes (**A**–**D**), known immune-related genes (**E**–**H**), signaling genes (**I**,**J**), vanadium binding protein genes (**K**,**L**), and transcription factor genes (**M**–**P**) were examined. Expression is shown relative to the reference gene (KY21.Chr2.148), which was identified by RNA-seq to be constantly expressed. Data are shown as the mean ± SEM with data points (white circles). Three to four independent data sets were analyzed by one-way ANOVA and *P_A_* values are indicated in each graph. Different alphabets are considered statistically significant (*p* < 0.05). The expression value of AMP genes for the 0 h was set to 1, given that the expression levels before CiEMa stimulation varied among the sample sets. N.E., not examined.

**Figure 9 ijms-25-01979-f009:**
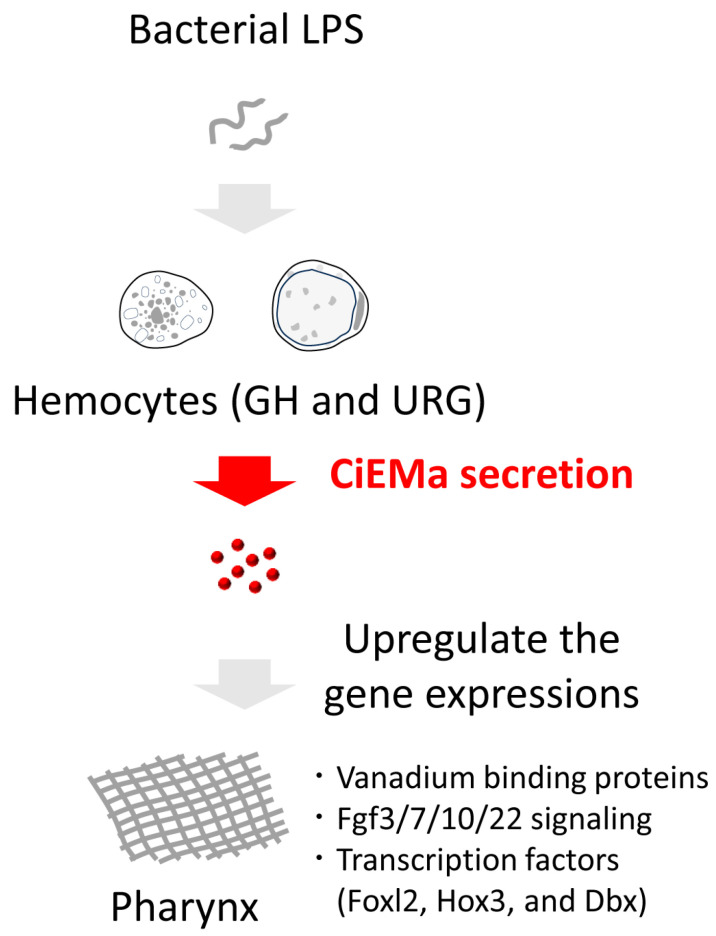
Schematic diagram summarizing the proposed model. CiEMa secretion from the hemocytes, namely GHs and URGs, is induced in response to bacterial lipopolysaccharide (LPS) as non-self-antigen. The CiEMa in turn affects the pharynx (indicated by the cross-hatched surface) and upregulates the expression of genes regarding vanabins, signaling molecules, and transcription factors. GH, granular hemocyte; URG, unilocular refractile granulocyte.

## Data Availability

The fastq files have been deposited in the NCBI database (PRJNA1045840).

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
