# Peer review of "A Novel Hemocyte-Derived Peptide and Its Possible Roles in Immune Response of Ciona intestinalis Type A"

_ijms, 2024, doi:10.3390/ijms25041979_

Round 1

Reviewer 1 Report

Comments and Suggestions for Authors

Peptides are small molecules but multi-diverse in structure and function. They have been purified from different natural sources. The functional analysis has revealed different roles mainly as antimicrobial molecules playing a major role in the immune response. However, less is known about hemocyte-derived peptides. For this reason, the authors have tried to decipher the role of a novel hemocyte-derived peptide in the in immune response of Ciona intestinalis type A.  Overall, the manuscript brings novel findings to the field, but the presentation of the results (graphs) must be carefully elaborated. The statistical analysis should be highlighted in the graphs. A major issue is the high number of self-citations. Below, you can find some suggestions and points to be clarified or addressed.

1. The abstract can be enriched with quantitative data.

2. Lines 28-29. Please clarify and specify what providing insights into the molecular and functional diversity of the immune response in chordates means. What are the insights?

3. Some sentences need to be supported by appropriate references. For example, lines 33-34, 34-37,

4. Please clarify the mean of molecular variation of the innate immune system

5. Lines 39-40. What is the difference between recognition and effector molecules? Provide examples.

6. The introduction is too long. The last paragraph basically repeats the information already presented in the abstract, results, discussion, and conclusion.

7. The objective of this research needs to be clearly stated in the introduction section.

8. The quality of Figures 1 and 2 can be improved.

9. Figure 6B. Please include statistical analysis.

10. I recommend using more technical software to draw the figures. The quality and the representation are not adequate.

11. Table 1 can be included as Supplementary Material. It was not explored and discussed in detail in the manuscript.

12. Figure 9. Identify graphs with letters. Include statistical analysis.

13. Lines 257-262. This is repeated in the discussion and conclusion.

14. Figure 10 should be included only in the discussion section. Please also improve the quality of the figure. I recommend working on a more elaborated diagram or including it as supplementary material. The manuscript contains an excessive number of figures.

15. Please do not cite the figures in the discussion section. They were previously mentioned in the results sections.  Sometimes authors repeat the same. I suggest focusing on discussing the results, not presenting them again.

16. The limitations of the study must be discussed.

17. The relevance and key contributions of this study need to be mentioned in the discussion.

18. Was the novel sequence of this peptide deposited in some database?

19. How stable is this peptide?

20. Figure 2 can be included as supplementary material. The main text must include only the main findings of the manuscript.

21. Figure 3. Individual images must be identified.

22. Please avoid excessive self-citations. Use only when is not possible to use another reference. I find a significant number of references from the authors. For example, 10, 11, 20, 24, 25, 26, 27, 28, 29, 36, 38 and 39. They correspond to about 18.2% of the references of this manuscript. 

Reviewer 2 Report

Comments and Suggestions for Authors

Major comments:

This article addresses an interesting topic with significant potential in the results obtained; however, the manuscript proves to be challenging to read in terms of maintaining continuity in its writing. In this sense, the Results section appears to be excessively lengthy, potentially due to the continuous inclusion of discussion elements within this section. It is recommended that the authors carefully review the manuscript to ensure that discussion content is excluded from the Results section. The Results section should be limited to the presentation of data and findings, allowing for a clear separation between results and discussion. As well, The Discussion section includes numerous specific references to results such as frequent references to figures. Authors are encouraged to focus on synthesizing the key findings and their implications. Clear and concise presentation of the data's significance will enhance the overall clarity and readability of the manuscript.

Minor comments: 

Lines 57-59:  The authors advance the existence of Ciona-specific mechanisms of immune response. However, it is essential to clarify why the authors suggest that the presence of uncharacterized peptides implies the existence of specific immune response mechanisms in Ciona. It should be noted that the citation number 21, to which these lines are referred, is pertinent to a methodology for the challenging task of certain AMP discovery.
